# Metabolic Profiling of the Oil of Sesame of the Egyptian Cultivar ‘Giza 32’ Employing LC-MS and Tandem MS-Based Untargeted Method

**DOI:** 10.3390/foods10020298

**Published:** 2021-02-02

**Authors:** Reham Hassan Mekky, Essam Abdel-Sattar, Antonio Segura-Carretero, María del Mar Contreras

**Affiliations:** 1Department of Pharmacognosy, Faculty of Pharmacy, Egyptian Russian University, Badr City, Cairo-Suez Road, Cairo 11829, Egypt; 2Research and Development Functional Food Centre (CIDAF), Bioregiόn Building, Health Science Technological Park, Avenida del Conocimiento s/n, 18016 Granada, Spain; ansegura@ugr.es; 3Department of Pharmacognosy, Faculty of Pharmacy, Cairo University, El Kasr El-Aini Street, Cairo 11562, Egypt; essam.abdelsattar@pharma.cu.edu.eg; 4Department of Analytical Chemistry, Faculty of Sciences, University of Granada, Avenida Fuentenueva s/n, 18071 Granada, Spain; 5Department of Chemical, Environmental and Materials Engineering, Campus Las Lagunillas, Universidad de Jaén, 23071 Jaén, Spain

**Keywords:** sesame oil, *Sesamum indicum* L., phenolic acids, lignans, flavonoids, core-shell columns, tandem MS

## Abstract

Sesame (*Sesamum indicum* L.) is a global oil crop. Sesame oil has been regarded as functional oil with antioxidant properties in several in vivo studies but little is known about its minor fraction. In this line, this study figures out the profile of the polar fraction of Egyptian cultivar Giza 32 sesame oil (SG32 oil) employing reversed-phase high-performance liquid chromatography coupled with diode array detection and electrospray ionization-quadrupole-time-of-flight-mass spectrometry and tandem MS. The characterization of the sesame oil metabolites depended on the observation of their retention time values, accurate MS, and MS/MS data, with UV spectra, and compared with relevant literature and available standards. Remarkably, 86 metabolites were characterized and sub-grouped into phenolic acids, lignans, flavonoids, nitrogenous compounds, and organic acids. From the characterized metabolites, 72 compounds were previously characterized in SG32 cake, which presented antioxidant properties, and hence it could contribute to SG32 oil antioxidant properties. Further studies are required to state the presence of such phenolics in commercial sesame oils and what of these compounds resist oil refining.

## 1. Introduction

Family Pedaliaceae belongs to the order Lamiales and consists of 14 genera and 70 species, including the famously known as sesame (*Sesamum indicum* L.) [1]. It is a crop-producing oil whose cultivation is distributed globally. On the base of total production, sesame seeds production is nearly 7 million tons with a production of 2 million tons of sesame oil [2]. Anciently, sesame originated from India. It was known in the Ancient Egyptian Civilization for the treatment of asthma since the third century BC [3,4,5]. Sesame seeds contain fats, proteins, carbohydrates, vitamins, and dietary fibers [5,6].

Besides their nutritional properties, several scientists explored possible biological activities and phytoconstituents of sesame seeds. In this sense, Dravie et al. [7] examined the antioxidant properties, total phenols, and flavonoids contents of Ghanaian sesame seeds via a multiple solvent extractions model. It was clear that the acetone extract was beyond most of the extracts in the antioxidant potential that were positively correlated with the total phenol content. As a matter of fact, sesame seeds have a myriad of biological activities, for instance, cardioprotective, hypolipidemic, hypocholesterolemic, anticancer, antioxidant, antidiabetic, and antihypertensive, among others [5,8,9,10,11]. Similarly, sesame seed oil has been recently regarded as functional oil with antioxidant properties [12,13]. In this line, Kim et al. [8] unraveled the auditory-protective effect of sesame oil in zebrafish and mice through the regulation of the hearing-related gene, Tecta. Concerning agri-food residues of sesame seeds, Khaleel et al. [14] have shown that residual sesame parts may exert some bioactivities.

Regarding the phytoconstituents retrieved from sesame, Dachtler et al. [15] characterized some furofurano lignans *viz*., sesamin, sesaminol, 6-episesaminol, and sesamolin using an online system of high-performance liquid chromatography (HPLC) hyphenated with nuclear magnetic resonance (NMR) and mass spectrometry (MS) detectors. In addition, Hassan [6] focused on the fatty acids composition of sesame oil. Moreover, Wu et al. [16] performed an LC-MS/MS combined with magnetic carboxylated multi-walled carbon nanotubes based-method for the determination of phenolic compounds in sesame oil. In another study, Görgüç et al. [17,18] incorporated vacuum and ultrasound-assisted enzymatic extraction to recover proteins and bioactive metabolites from sesame bran. Recently, Mekky et al. [10] performed untargeted metabolic profiling of sesame cake via reversed-phase (RP) (HPLC)—diode array detection (DAD) and hyphenated to electrospray ionization (ESI)-quadrupole-time-of-flight (QTOF)-MS and tandem MS. This technology enables the detection of phenolic acids, lignans, and flavonoids, giving an insight into the significance of such agri-food residue.

Similarly, this study aims at performing untargeted profiling of the metabolites of the polar fraction of sesame oil of the Egyptian cultivar ’Giza 32’ (SG32) through RP-HPLC-DAD-QTOF-MS and tandem MS to give new insights into the minor composition, with a focus on phenolic compounds. In fact, there is little information concerning the metabolic profiling of sesame oil, which has been limited to some phenolic classes and fatty acids, as commented before.

## 2. Materials and Methods

### 2.1. Chemicals

Solvents (*n*-hexane, methanol, acetonitrile, acetone, and glacial acetic acid) were obtained from Fisher Chemicals (Thermo Fisher Scientific, Waltham, MA, USA). They were of analytical and MS grade for extraction and characterization, respectively. A Milli-Q system (Millipore, Bedford, MA, USA) was used for obtaining ultrapure water. Standards were purchased from Sigma-Aldrich (St. Louis, MO, USA), except for some amino acids (L-tryptophan and L-phenylalanine), which were from Acros Organics (Morris Plains, NJ, USA). The degree of purity of all the used standards was around 95% (*w*/*w*).

### 2.2. Samples Procurement and Oil Extraction Procedures

SG32 seeds were identified and provided by Agriculture Engineer Nadia Abdel-Azim, Egyptian Ministry of Agriculture and Land Reclamation (Giza, Egypt). Firstly, they were ground into a fine powder with a particle size of around 1 mm via an Ultra Centrifugal Mill ZM 200, Retsch (Haan, Germany).

The extraction of SG32 seeds was according to Shyu and Hwang [19], with some modifications. The first step was to extract sesame oil through the homogenization of 1 g of SG32 seeds with 10 mL *n*-hexane utilizing a magnetic stirrer Agimatic-N (Jp Selecta, Barcelona, Spain) for 30 min at room temperature followed by centrifugation at 7155× *g* at 5 °C for 15 min by a Sorvall ST 16 (Thermo Sci., ThermoFisher, Waltham, MA, USA). To recover oil fraction from SG32 seeds, *n*-hexane supernatant was collected and evaporated until dry under vacuum by a rotary evaporator at 38 °C (Rotavapor R-200, Büchi Labortechnik, AG, Switzerland). As the second step and in order to extract the phenolic compounds from sesame oil, this fraction was reconstituted in 2 mL *n*-hexane followed by 5 mL methanol:water (80:20, *v*/*v*) based on Ishtiaque et al. [20] to get the polar fraction of SG32 oil. The extraction mixture was agitated, centrifuged, and the supernatant collected. The previous step was repeated twice (methanol:water (80:20, *v*/*v*) × 2.5 mL). The methanolic-aqueous extracts were combined and defatted with *n*-hexane (2 mL) to eliminate any residual fat and concentrated using a speed-vacuum Concentrator plus (Eppendorf AG, Hamburg, Germany) at 30 °C for above 3 h. The polar oil fraction was appropriately dissolved in 2 mL of aqueous methanol (80:20, *v*/*v*) previously to subjection to RP-HPLC-DAD-ESI-QTOF-MS and -tandem MS analysis.

### 2.3. Analysis by RP-HPLC-DAD-ESI-QTOF-MS and -Tandem MS

The HPLC was an Agilent 1200 series equipped with a binary pump, an autosampler, and a diode array detector (DAD), (Agilent Technologies, Santa Clara, CA, USA) [21,22]. The separating column was a core-shell Halo C18 (150 mm × 4.6 mm, 2.7 μm particle size, Advanced Materials Technologies, Wilmington, DE, USA). The system was hyphenated to a 6540 Agilent Ultra-High-Definition (UHD) Accurate-Mass Q-TOF LC/MS equipped with an Agilent Dual Jet Stream electrospray ionization (Dual AJS ESI) interface. MassHunter Workstation software (Agilent Technologies) was used for data acquisition (2.5 Hz) in profile mode. The spectra were acquired over a mass-to-charge (*m/z*) range from 70 to 1500 in negative-ion mode. The detection window was set to 100 ppm.

Data analysis was performed using MassHunter Qualitative Analysis B.06.00 (Agilent Technologies) according to [10,21,23,24]. In brief, compounds characterization was performed by observing candidates and the generation of their formulas within a mass error limit of ±5 ppm. The MS score was set to ≥80 [10]. The following databases were consulted: Reaxys [25],KNApSAcK Core System [26], SciFinder Scholar [27], PubChem [28], ChemSpider [29], METLIN Metabolite Database [30], Phenol-Explorer [31], the Dictionary of Natural Products [32], and Phytochemical dictionary of natural products database [33]. Moreover, relevant literature was traced via the Egyptian Knowledge Bank [34]. In addition, for characterization work validation, a comparison was made with standards whenever possible.

## 3. Results and Discussion

### 3.1. RP-HPLC-DAD-ESI-QTOF-MS and Tandem-MS of SG32 Oil

The phenolic fraction of SG32 oil was analyzed in negative ionization mode via core-shell RP-HPLC-DAD-ESI-QTOF-MS and tandem MS. Table 1 and Table 2 classify the characterized metabolites into phenolics and non-phenolics. Besides, they demonstrate for each candidate the time (RT), experimental *m*/*z*, generated molecular formulas, mass errors, scores, double bond equivalents (DBE), UV maxima (if present), tandem mass fragments, and relative abundance (area of chromatographic profiles of all characterized metabolites), respectively. This information was used for the characterization work, which was based on the strategy followed in our previous studies. Basically, the RT, molecular formula, and the fragmentation patterns were compared to those found in literature, databases, and standards, when possible. Moreover, fragmentation patterns enabled us to obtain clues about the functional groups, basic constituents, and/or polyphenol nucleus [10,23,35]. Nonetheless, further confirmation is required by NMR to also establish the stereochemistry.

Also, Appendix A (Appendix A) mention metabolites classifications, and cite their previous description in the literature. A total of 86 metabolites were characterized with 11 metabolites reported for the first time in sesame, 59 metabolites observed for the first time in sesame oil, and 3 new proposed structures. 

Moreover, Figure 1a represents the base peak chromatogram obtained by RP-HPLC-DAD-ESI-QTOF-MS showing the complexity of the minor constituents of sesame oil as that for the seed cake (Figure 1c).

#### 3.1.1. Phenolic Compounds

Sixty-four phenolic compounds were observed in SG32 oil and could be categorized into phenolic acids (32), flavonoids (19), lignans (10), coumarins (1), phenol aldehydes (1), and phenol derivatives (1). Figure 1b summarizes these compounds grouped into these phenolic classes, showing their position in the RP-HPLC-MS chromatogram and their abundance.

#### Phenolic Acids

The presence of phenolic acids was noticed with 32 phenolic compounds, being the major class of the annotated metabolites in qualitative and quantitative terms (Figure 1b, see Section 3.2). They belonged to three subclasses *viz.*, hydroxybenzoic acids (12), hydroxycinnamic acids (19), and a hexahydroxydiphenic acid dilactone (Table 1 and Appendix A).

Concerning hydroxybenzoic acids, compounds at *m/z* 121.03 and 135.05 showed the loss of CO (28 Da) and CO_2_ (44 Da), and λ_max_ 278 and 273 nm, respectively. They were described as benzoic acid and a methyl derivative [10]. Methylated and methoxy derivatives were also tentatively identified. In this sense, compounds at *m/z* 151.40 (C_8_H_8_O_3_) exerted neutral losses of methyl (CH_3_, 15 Da) followed by decarboxylation (CO_2_, 44 Da), which are typical of the presence of methoxy groups and phenolic acids, respectively. They were annotated as methoxybenzoic acid isomers I–II according to the Reaxys database. Figure 2a describes the main fragments of methoxybenzoic acid isomer II. Similarly, a methylated derivative was observed at RT 14.61 min and was characterized as hydroxybenzoic acid methyl ester. In this case, a loss of CH_2_ was observed instead of CH_3_ as in the aforementioned cases. Moreover, a mono-hydroxylated benzoic acid was also characterized, hydroxybenzoic acid (*m/z* 137.02, C_7_H_6_O_3_), with the sequential loss of water and CO_2_ [21].

Dihydroxybenzoic acids were detected, and illustrated as, protocatechuic and vanillic acids (*O*-methylated derivative), at RT 11.26 and 16.79 min, respectively. Both revealed the neutral loss of water (18 Da) and CO_2_ with an additional loss of a methyl group (CH_3_) in the case of vanillic acid [36] (Figure 2b). A glycosylated derivative of this compound (vanillic acid hexoside) was also characterized (*m/z* of 329.09, C_14_H_18_O_9_), which showed a loss of a hexose (162 Da) as being linked through the hydroxyl moiety of the vanillic acid, as well as CH_3_ and CO_2_, as for the aforementioned phenolic acids. It bears noting that this is the first report of it in sesame oil [10].

Gallic and syringic acids presence (tri-hydroxylated benzoic acids) was confirmed with standards. Moreover, syringic acid hexoside was detected at (RT 11.81 min, *m/z* 359.10) showing the neutral loss of a hexose, demethylation, and decarboxylation, as before [10] (Table 1 and Appendix A).

Concerning hydroxycinnamic acids, 19 derivatives were observed. They could be divided into cinnamic acid (non-hydroxylated), *p*-coumaric acid, and *m*-coumaric acid derivatives (mono-hydroxylated), caffeic acid, and ferulic acid derivatives (di-hydroxylated), and sinapic acid derivatives (tri-hydroxylated). It is noteworthy that the presence of *m*-coumaric, *p*-coumaric acid, chlorogenic (caffeoylquinic I), and ferulic acids was unambiguously confirmed with standards. Another isomer of caffeoylquinic acid was observed at RT 17.23 min with the presence of the fragment ions of quinic acid with its dehydrated ion (*m/z* 191.0556 and 173.0450) and caffeic acid with its dehydrated and decarboxylated ions (*m/z* 179.0327, 161.0233, and 135.0455) [24]. Besides, the occurrence of *m*-coumaric acid and caffeoylquinic acids I–II is observed for the first time in sesame oil. In this line, a caffeoyl phenylethanoid derivative (*m/z* 623.20, C_29_H_36_O_15_) was observed at RT 21.12 min. The main detected fragments unraveled the neutral loss of a caffeoyl and a deoxyhexosyl moieties (*m/z* 461.17 and 315.11) with the detection of caffeic acid ion and its dehydrated form (*m/z* 179.03 and 161.02). The hydroxytyrosol ion (phenylethanoid) was observed (*m/z* 153.05) after the neutral loss of hexosyl moiety (Figure 3a) and so it is verbascoside according to previous studies [10,24].

Additionally, peak 21 (*m/z* of 147.05, C_9_H_8_O_2_) showed the characteristic neutral loss of 44 Da of phenolic acids and thus it was annotated as cinnamic acid [10] (Table 1, Appendix A). Four isomers of *p*-coumaric acid hexosides were characterized at *m/z* values of 325.09 (C_15_H_18_O_8_) and with MS/MS revealing neutral losses of hexose, which release the aglycone nuclei (*m/z* 163.04), followed by decarboxylation. Similar fragmentation patterns were obtained for four isomers of ferulic acid hexoside (peaks 30, 31, 39, and 45) [10]. Concerning tri-hydroxylated cinnamic acids, four sinapic acid derivatives were firstly detected in sesame oil. Briefly, sinapic acid hexoside was observed (*m/z* 385.11, C_17_H_22_O_10_) with neutral loss of the hexosyl moiety (162 Da) followed by the common fragments of sinapic acid [10]. In the same manner, sinapic acid deoxyhexoside hexoside was annotated at RT 17.83 min with the aforementioned fragmentation pattern and the additional loss of deoxyhexose. Finally, two undescribed sinapoyl-dehydroshikimic acid hexosides were detected, showing the neutral loss of a hexose (*m/z* 377.12) with subsequent sinapic acid ion (*m/z* 223.06) and the sequential loss of a methyl group (*m/z* 209.0455) and water (*m/z* 191.03). Moreover, dehydroshikmic acid ion was detected with *m/z* 171.03 followed by its dehydrated (*m/z* 153.0537) and decarboxylated (*m/z* 127.04) forms (Table 1 and Appendix A), Figure 3b shows the detailed fragmentation pattern.

In line with hexahydroxydiphenic acid dilactone, ellagic acid was observed upon comparison with a standard at RT 20.93 min.

#### Flavonoids

The presence of flavonoids in SG32 oil was widely observed with 19 derivatives. They are sub-grouped into flavonols (5), flavones (10), a flavan-3-ol, a proanthocyanidin, and flavanones (2) (Table 1). It is noteworthy that the detection of kaempferol 3-*O*-β-D-glucopyranoside, kaempferol 3-*O*-rutinoside, quercetin 3-*O*-β-D-glucopyranoside, quercetin 3-*O*-rhamnopyranoside, quercetin 3-*O*-rutinoside, luteolin, luteolin 7-*O*-β-D-glucopyranoside, (-)-epicatechin, procyanidin A2, and naringenin was based on standards comparison. They were all mentioned for the first time in sesame oil, while procyanidin A2 was also described for the first time in Pedaliaceae. This compound is found in other families like Ericaceae [37]. Figure 4a,b shows examples of the fragmentation patterns, highlighting the typical losses of hexose of *O*-glycosylated compounds.

A similar fragmentation pattern to quercetin derivatives was observed for hesperetin hexoside deoxyhexoside (RT 23.18 min, *m/z* 609.18, C_28_H_34_O_15_), with sequential losses of a hexosyl (162 Da, *m/z* 447.13) and a deoxyhexosyl (146 Da, *m/z* 301.07) from the *O*-hexoside and deoxyhexoside moieties. The fragmentation pattern of the aglycone unraveled the common fragment ion of *m/z* 259.08 for (M-H-CH_2_CO) followed by the ions at *m/z* 175.00 (^0,4^B^−^) and 151.00 (^1,3^A^−^) [38]. Furthermore, UV absorbance λ_max_ 281 nm indicated a flavanone structure. As far as we know, it is the first description of it in Pedaliaceae [39].

Mostly, flavones were represented in SG32 oil as *C*-glycosides conjugates of either apigenin or luteolin with the typical fragmentation of *C*-glycosides. This is characterized by the loss of 18 Da (H_2_O), 44 Da (CO_2_), 60 Da (2 × (CH_2_O)), 90 Da (3 × (CH_2_O)), and/or 120 Da (4 × (CH_2_O)), according to previously reported studies [10,40]. In this line, apigenin *C*-pentoside *C*-hexoside (I–III) isomers were noticed exerting fragments at *m/z* values of 541.13, 503.12, 473.11, 443.10, 383.08, and 353.07, and with the common ion at *m/z* 117.03 (^1,3^B^−^), Figure 4c [21,38,39,41]. As for luteolin derivatives, two isomers of luteolin *C*-hexoside I–II and luteolin *C*-deoxyhexoside-*C*-hexoside I–II were annotated being characterized by a similar fragmentation pattern to the aforementioned apigenin derivatives and comparison with reported studies [10,42]. To our knowledge, this is the first report of apigenin and luteolin derivatives in sesame oil. Another luteolin derivative, but *O*-glycosylated, was observed at *m/z* 593.15 (C_27_H_30_O_15_) and thus it showed the neutral loss of a deoxyhexosyl (*m/z* 447.09) and a hexosyl (*m/z* 285.04) moieties. The aglycone also revealed fragment 133.03 suggesting the ion (^1,3^B^−^) and hence was described as luteolin deoxyhexoside hexoside, which was not reported before in genus *Sesamum*, as far as we know.

#### Lignans

Lignans are dimeric *β-β’*-linked phenylpropanoid compounds that are widely distributed in Kingdom Plantae and possess several biological activities [43]. Ten lignan derivatives were observed in SG32 oil. All of them are classified as furofuran lignans and they occurred as sugars conjugates where the loss of sugars was observed and the aglycones analogs were compared with previously reported studies [10,43,44]. Concisely, two isomers of pinoresinol dihexoside were annotated exerting the neutral loss of two hexosyl moieties (*m/z* 357.1302, 2 × 162 Da) with an aglycone fragmentation showing an *m/z* of 151 resulted from the cleavage of the tetrahydrofuran ring complying with earlier reports [10,43,44,45] (Table 1 and Appendix A). Similarly, the ion *m/z* of 841.28 (RT 22.52 min, C_38_H_50_O_21_.) exhibited a neutral loss of three hexosyl moieties consecutively, leading the fragment ions *m/z* 679.22, 458.15, and 355.12). Besides, the fragments *m/z* 161 and *m/z* 149 were noticed standing for 1,3-dioxymethylenephenyl-CHCHCH_2_ and 1,3-dioxymethylenephenyl-CO, respectively [44]. Consequently, it was tentatively characterized as xanthoxylol trihexoside. Correspondingly, the unreported analog xanthoxylol dihexoside was observed at RT 24.46 min with a similar fragmentation pattern (Table 1 and Appendix A, Figure 5a).

About sesaminol, the presence of sesaminol trihexoside (I–III) and sesaminol tetrahexoside (I–II) isomers was noticed. As before, they were characterized by the corresponding losses of hexosyl moieties with the appearance of sesaminol aglycone (*m/z* 369.10). Besides, the latter aglycone exhibited the fragments of *m/z* 161 and *m/z* 149 complying with furofurano lignans [10,46]. Moreover, hydroxysesaminol trihexoside was detected as being characterized by the ion of the aglycone *m/z* 385 with the latter characteristic ion at *m/z* 161 (Figure 5b). All these lignans glycosides have been observed for the first time in sesame oil.

#### Coumarins, Phenol Aldehydes, and Derivatives

Besides the aforementioned classes, the coumarin 7-hydroxycoumarin (umbelliferone) presence was unambiguously confirmed upon comparison with a standard, while sesamol (*m/z* 137.02) [10] and vanillin (phenol aldehyde) were tentatively characterized [47].

#### 3.1.2. Non-Phenolic Compounds

##### Nitrogenous Compounds

Concerning nitrogenous compounds, the occurrence of amino acids was observed by eight derivatives namely pyroglutamic acid (I–II) leucine/isoleucine (I–III), tyrosine, phenylalanine, and tryptophan. They exhibited the neutral loss of ammonia (17 Da) and/or CO_2_ (44 Da) complying with several reports [10,21,24,36,45]. Besides, the aromatic amino acids phenylalanine, tyrosine, and tryptophan were confirmed with standards. It bears noting that this is the first report of pyroglutamic acid in Pedaliaceae (Table 2 and Appendix A). In this line, oxidized glutathione (GSSG) was detected (RT 6.32 min, *m/z* 611.14) showing both glutathione (*m/z* 306.08) and glutamyl (*m/z* 128.04) moieties [10] and hence giving a clue of the occurrence of reduced glutathione (GSH) in SG32 oil being susceptible to conversion to GSSG during sampling and analysis [48]. As a matter of fact, GSH is a natural cellular antioxidant that prevents the onset and progression of many serious diseases [48]. This is the first report of GSSG in sesame oil and its presence in sesame oil provides a new aspect of its functionality (Table 2 and Appendix A).

##### Organic Acids

Thirteen organic acids were detected in SG32 oil *viz.*, gluconic/galactonic, citric (I–III), malic (I–II), succinic, (−)-3-dehydroshikimic, quinic (I–II), pantothenic, isopropylmalic, and azelaic acids. Their characterization are according to earlier reports [10,24,36,47,49,50]. All of them, as far as we know, were not reported before in sesame oil (Table 2 and Appendix A).

### 3.2. Semi-Quantitative Analysis

Semi-quantitative analysis was performed via estimation of the total peak area obtained by MS denoting the relative amount of each characterized metabolite. In the perspective of subclasses of phenolic metabolites, phenolic acids subclass was the most abundant with (38.4%) followed by flavonoids (33.7%) then lignans (26.9%) (Figure 6).

Concerning individual phenolic compounds, sesaminol trihexoside III (18%), followed by sinapoyl-3-dehydroshikimic acid hexoside I (10%), apigenin *C*-pentoside *C*-hexoside I (7%), methyl benzoic acid (5%), and apigenin *C*-pentoside *C*-hexoside III (5%) were the most abundant phenolic metabolites. In this context, *C*-glycosides are considered antioxidant capacity enhancers as the hydroxyl group and metal chelation sites in flavones are free [24,41] and so it is expected that these phenolic compounds can protect the oil from oxidation.

### 3.3. Comparison between Sesame Seed Oil and Cake

Our previous study on SG32 cake counterpart exhibited the presence of 112 metabolites [10]. They belonged to the same classes of phytoconstituents in SG32 oil. Remarkably, 72 metabolites were detected in both SG32 cake and oil (Appendix A). The remaining metabolites in SG32 oil consist of generally minor phenolic compounds, and no sugars were observed as in the cake [10]. All the characterized lignans were of furofurano type, as in the seed cake. GSSG was detected in both samples.

Basically, sesame oil is considered a functional oil [12,13]. Since SG32 cake showed antioxidant activities, the common phenolic compounds between SG32 cake and oil could contribute to the biological potential of the oil. The phenolic profile in vegetable oils depends on the source and the processing. For example, olive oil is rich in hydroxytyrosol derivatives, among other phenolic compounds. Some of these compounds come from the olive fruit and pass to the oil, which is obtained by milling, malaxation, and centrifugation [51]. Nonetheless, the processing conditions affect the profile and the content of phenolic compounds in olive oil and thus its oxidative stability and shelf life [52,53]. Another example is tea seed oil obtained by screw pressing extraction, which also present tea phenolic compounds [54] and they contribute to the antioxidant stability [55]. In this context, it is important to establish the phenolic profile in vegetable oils due to the contribution of phenolic compounds to the functional and antioxidative properties of the vegetable oils. Therefore, other studies should be performed to address the phenolic composition integrity of virgin cold-pressed sesame oil and refined oil as industrial processes could lead to the loss or modification of phenolics compounds [56,57]. The employment of the state of art hyphenated techniques as U/HPLC, like in the current work, and/or GC coupled to high-resolution MS could help in this purpose as other authors have shown for other vegetable oils [51,53,54,58]. Even more, the data obtained here can be the basis for those characterization studies on sesame oils when using U/HPLC-MS. In fact, lignan aglycones have not been found in this work. It seems that lignans can be transformed during the oil production process, and some of them, like sesamolin and sesamin, are unstable [59].

## 4. Conclusions

In the present study, core-shell RP-HPLC–DAD–ESI–QTOF-MS and -MS/MS were utilized for the analysis of the oil of the Egyptian cultivar of sesame ′Giza 32’. Collectively, 86 metabolites were characterized in sesame SG32 oil, with 64 phenolic compounds with 3 unreported metabolites. The observed phenolic compounds were classified into phenolic acids, flavonoids, lignans, and others. All the characterized lignans were of furofurano type, as in the seed cake. Moreover, this is the first report showing oxidized glutathione in sesame oil. Mostly, the phenolic metabolites and other phytoconstituents in SG32 oil were present in SG32 cake counterpart, suggesting that they can pass from the seed cake to the oil, while in some cases the oil counterpart seems to be enriched. Consequently, further studies are required to detect the presence of important bioactive metabolites in commercial samples.

## Figures and Tables

**Figure 1 foods-10-00298-f001:**
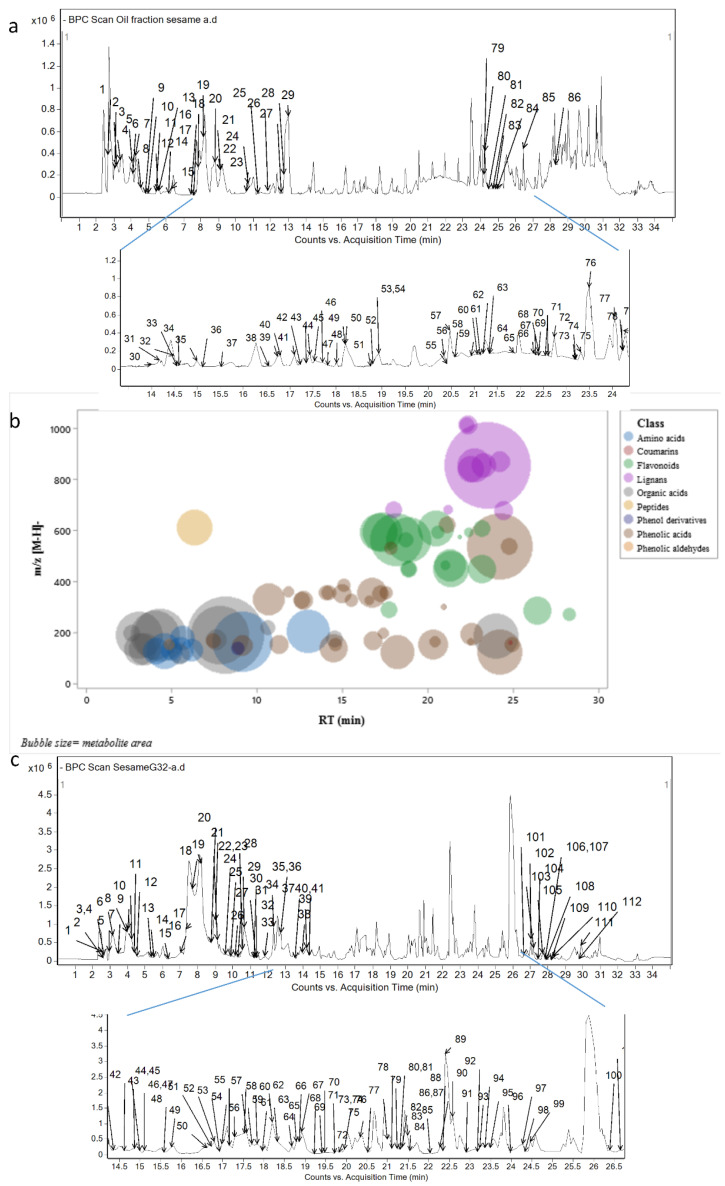
(**a**) Base peak chromatogram of SG32 oil and (**b**) its metabolites grouped into classes and classified according to *m/z*, retention time (RT), and relative area in SG32 oil. (**c**) Base peak chromatogram of SG32 cake, which has been adapted from [10].

**Figure 2 foods-10-00298-f002:**
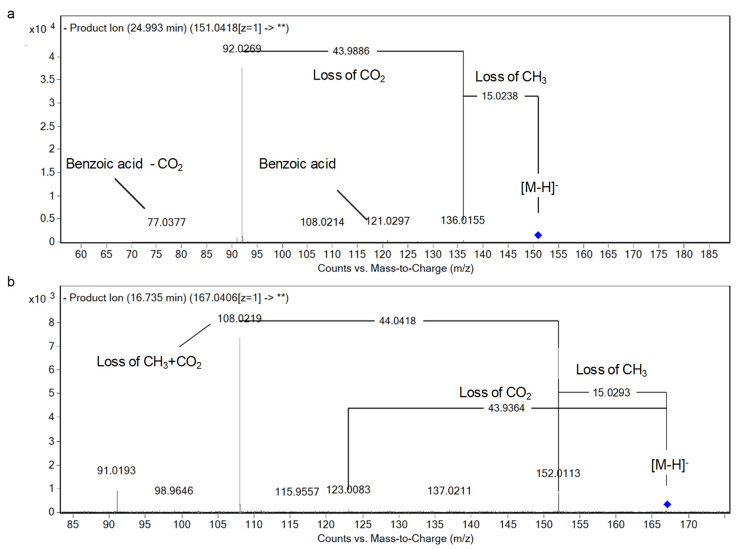
The patterns of fragmentation of (**a**) methoxybenzoic acid II and (**b**) vanillic acid.

**Figure 3 foods-10-00298-f003:**
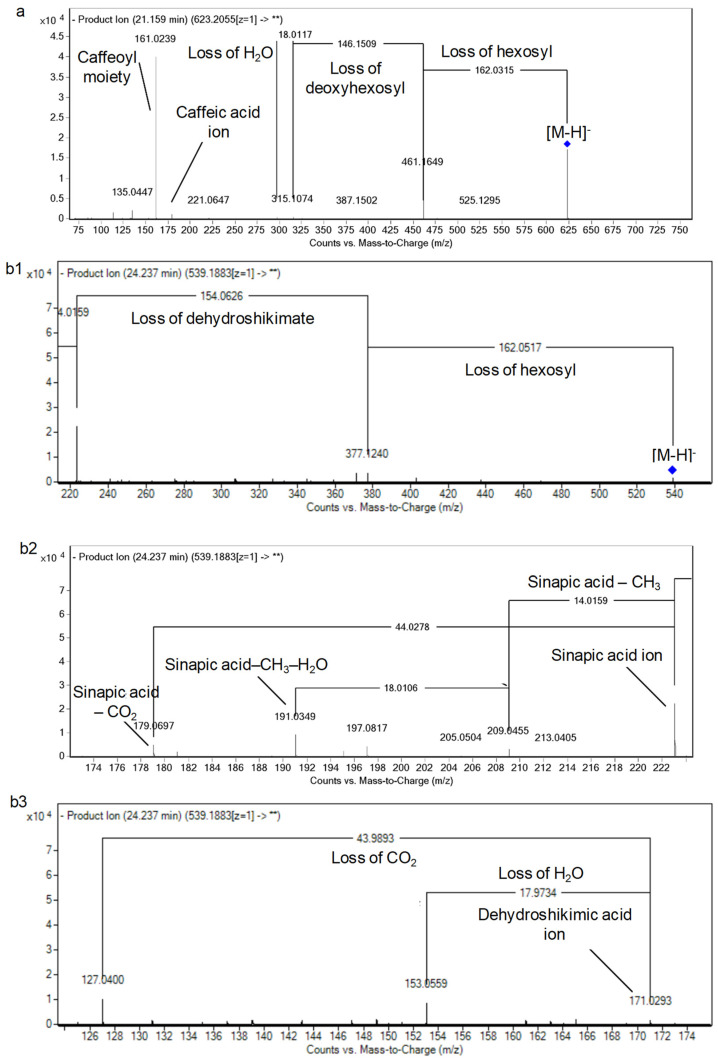
The patterns of fragmentation of (**a**) verbascoside, (**b**) sinapoyl-3-dehydroshikimic acid hexoside I. This spectrum has been divided (**b**1–**b**3) for a better description of the fragments.

**Figure 4 foods-10-00298-f004:**
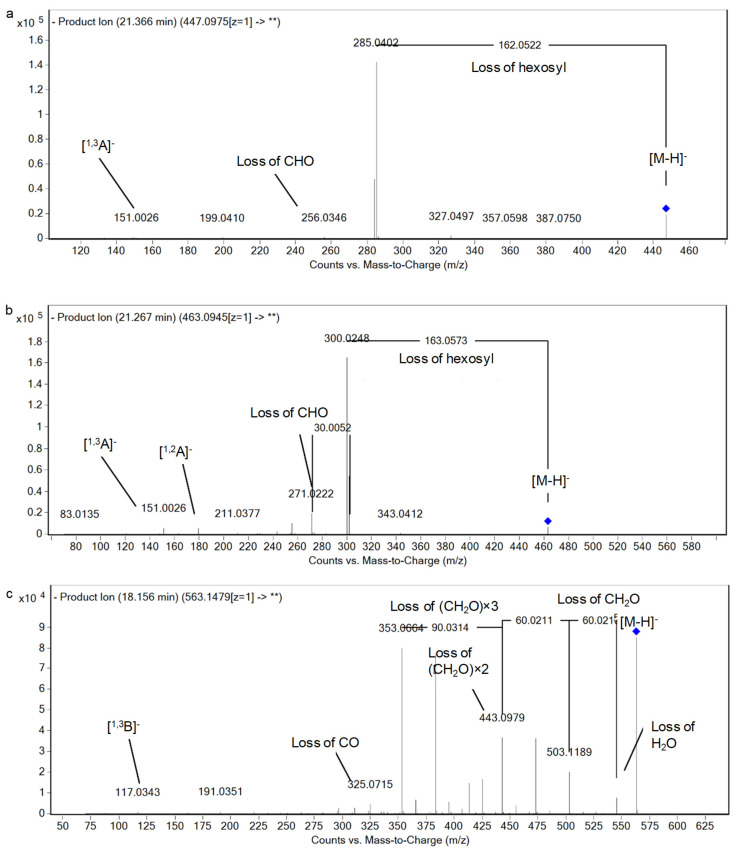
The patterns of fragmentation of (**a**) luteolin 7-*O*-β-D-glucopyranoside, (**b**) quercetin 3-*O*-β-D-glucopyranoside, and (**c**) apigenin *C*-pentoside C-hexoside I.

**Figure 5 foods-10-00298-f005:**
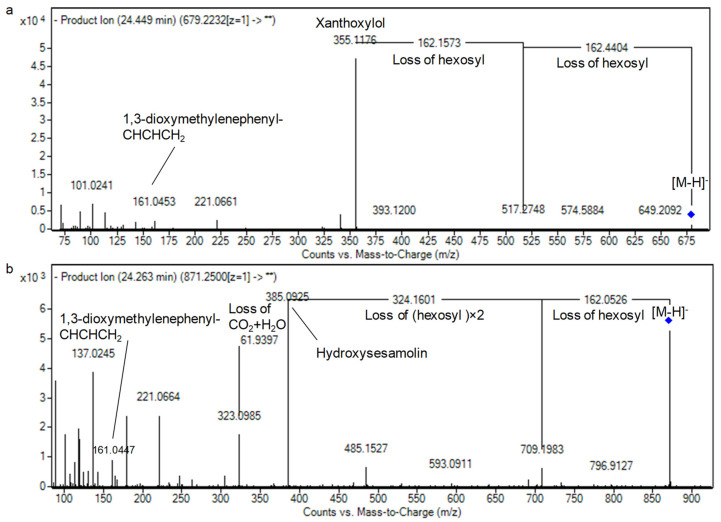
The patterns of fragmentation of (**a**) xanthoxylol dihexoside, and (**b**) hydroxysesamolin trihexoside.

**Figure 6 foods-10-00298-f006:**
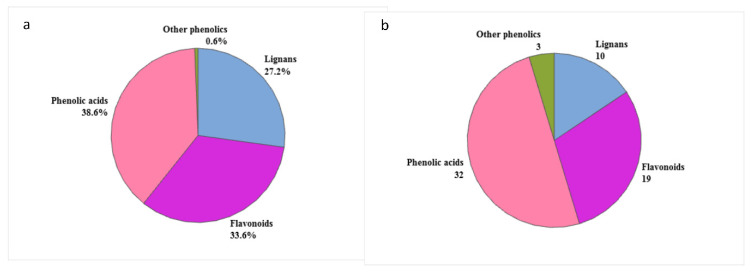
Phenolic compounds characterized in SG32 oil: (**a**) Relative abundance (%) and (**b**) qualitative classification (%).

**Table 1 foods-10-00298-t001:** Phenolic compounds characterized in SG32 oil.

Peak No.	RT (min)	Experimental *m/z* ^a^ [M-H]^−^	Theoretical Mass (M)	Molecular Formula	Score	Error (ppm)	Error (mDa)	Main Fragments	DBE	UV (nm)	Proposed Compound	Area (Response × RT)	%
9	4.7	151.0402	152.0473	C_8_H_8_O_3_	98.66	−0.3	0	123.0450, 122.0372	5	267	Vanillin	1.14 × 10^5^	0.26%
16	7.31	169.014	170.0215	C_7_H_6_O_5_	86.57	1.25	0.2	125.0247	5	N.D.	Gallic acid *	2.25 × 10^5^	0.51%
20	8.74	137.0243	138.0316	C_7_H_6_O_3_	94.24	0.34	0.05	N.D.	5	N.D.	Sesamol	1.43 × 10^5^	0.33%
21	9.19	147.0447	148.0524	C_9_H_8_O_2_	96.29	3	0.5	103.0556	6	N.D.	Cinnamic acid	3.97 × 10^5^	0.90%
24	10.64	329.0879	330.0951	C_14_H_18_O_9_	99.34	−0.2	−0.1	167.0347, 152.0114, 123.0450. 108.0218	6	255, 286	Vanillic acid hexoside	1.06 × 10^6^	2.40%
25	11.33	153.0192	154.0266	C_7_H_6_O_4_	98.75	1.02	0.16	137.0244, 109.0293, 108.0213	5	258, 288sh	Protocatechuic acid *	3.27 × 10^5^	0.74%
26	11.826	359.0978	360.1057	C_15_H_20_O_10_	96.98	1.2	0.4	197.0474, 182.0213, 153.0544, 138.0318	6	N.D.	Syringic acid hexoside	1.18 × 10^5^	0.27%
27	12.47	325.0932	326.1002	C_15_H_18_O_8_	93.93	0.4	0.1	163.0398, 119.0502	7	287	*p*-Coumaric acid hexoside I	2.86 × 10^5^	0.65%
28	12.72	325.0931	326.1002	C_15_H_18_O_8_	99.12	−0.5	−0.2	163.0400, 119.0502	7	288	*p*-Coumaric acid hexoside II	3.28 × 10^5^	0.74%
30	13.95	355.1035	356.1107	C_16_H_20_O_9_	99.4	−0.1	−0.03	193.0505, 178.0271, 149.0607	7	279	Ferulic acid hexoside I	2.31 × 10^5^	0.52%
31	14.01	355.1037	356.1107	C_16_H_20_O_9_	99.5	−0.7	−0.25	N.D.	7	282	Ferulic acid hexoside II	2.31 × 10^5^	0.52%
32	14.36	137.024	138.0317	C_7_H_6_O_3_	99.24	−1.1	−0.2	119.0142, 109.0293, 108.0218, 93.0344, 92.0269	5	273	Hydroxybenzoic acid *	7.59 × 10^5^	1.72%
33	14.49	151.0401	152.0473	C_8_H_8_O_3_	97.18	−0.9	−0.1	137.0245, 107.0506	5	237, 274	Hydroxybenzoic acid methyl ester	1.52 × 10^5^	0.34%
35	14.99	353.0879	354.0951	C_16_H_18_O_9_	99.38	−0.5	−0.2	191.0559,179.0344, 173.0457, 161.0246, 135.0452	8	238, 279	Caffeoylquinic acid I * (chlorogenic acid)	3.75 × 10^5^	0.85%
36	15.1	385.1137	386.1213	C_17_H_22_O_10_	87.7	−0.6	−0.23	223.0611, 208.0373, 193.0141, 179.0703, 164.0472, 149.0242	7	290	Sinapic acid hexoside	1.38 × 10^5^	0.31%
37	15.53	325.0928	326.1002	C_15_H_18_O_8_	97.52	−0.2	−0.1	163.0397, 119.0503	7	277	*p*-Coumaric acid hexoside III	1.60 × 10^5^	0.36%
38	16.61	325.0929	326.1002	C_15_H_18_O_8_	89.25	−1.1	−0.4	N.D.	7	N.D.	*p*-Coumaric acid hexoside IV	9.37 × 10^4^	0.21%
39	16.68	355.1033	356.1107	C_16_H_20_O_9_	99.7	0.5	0.18	193.0500, 178.0272, 149.0609	7	282	Ferulic acid hexoside III	9.21 × 10^5^	2.09%
40	16.73	167.0346	168.0426	C_8_H_8_O_4_	97.64	1.29	0.22	152.0144, 123.0449, 108.0251	5	258	Vanillic acid *	3.11 × 10^5^	0.71%
41	17.11	593.151	594.1585	C_27_H_30_O_15_	99.3	0.3	0.15	503.1188, 473.1085, 443.0981, 413.0872, 383.0767, 353.0667, 135.0457, 119.0357, 117.0367	13	267, 320	Luteolin *C*-deoxyhexoside *C*-hexoside I	1.42 × 10^6^	3.22%
42	17.23	353.0876	354.0951	C_16_H_18_O_9_	96.29	0.3	0.1	191.0556, 179.0327, 173.0450, 161.0233, 135.0445	8	274	Caffeoylquinic acid II	2.52 × 10^5^	0.57%
43	17.36	593.159	594.1585	C_27_H_30_O_15_	99.3	0.3	0.15	533.1285, 503.1193, 473.1085, 443.0989, 413.0873, 383.0771, 353.0665	13	270, 325	Luteolin *C*-deoxyhexoside *C*-hexoside II	1.59 × 10^6^	3.60%
44	17.34	197.0453	198.05282	C_9_H_10_O_5_	96.97	0.6	0.1	N.D.	5		Syringic acid *	1.12 × 10^5^	0.25%
45	17.47	355.1034	356.1107	C_16_H_20_O_9_	96.4	0.4	0.16	N.D.	7	287	Ferulic acid hexoside IV	1.64 × 10^5^	0.37%
46	17.78	289.0718	290.079	C_15_H_14_O_6_	92.99	0.79	0.02	253.0334, 245.1390, 217.0027, 131.0712, 123.0450	9	N.D.	(−)-Epicatechin *	2.52 × 10^5^	0.57%
47	17.78	531.1719	532.1792	C_23_H_32_O_14_	89.5	−0.5	−0.28	179.0140, 165.0554, 150.0317	8	283	Sinapic acid deoxyhexoside hexoside	1.46 × 10^5^	0.33%
48	17.96	681.24	682.2473	C_32_H_42_O_16_	99.63	0.54	0.37	357.1302, 151.0384	12	242, 275	Pinoresinol dihexoside I	2.96 × 10^5^	0.67%
49	18.21	563.1411	564.1479	C_26_H_28_O_14_	99.03	−0.57	−0.32	545.1297, 503.1189, 473.1087, 443.0979, 413.0872, 383.0771, 353.0664, 117.0343	13	274, 330	Apigenin *C*-pentoside *C*-hexoside I	2.93 × 10^6^	6.64%
50	18.27	121.0293	122.0368	C_7_H_6_O_2_	99.78	1.2	0.15	92.0269	5	278	Benzoic acid	1.22 × 10^6^	2.76%
51	18.63	563.1405	564.1479	C_26_H_28_O_14_	98.83	0.08	0.05	545.1323, 503.1179, 473.1083, 443.0976, 413.0871, 383.0767, 353.0666, 117.0335	13	272, 327	Apigenin *C*-pentoside *C*-hexoside II	2.20 × 10^5^	0.50%
52	18.89	563.1408	564.1479	C_26_H_28_O_14_	98.83	0.08	0.05	545.1298, 503.1191, 473.1087, 443.0976, 413.0878, 383.0769, 353.0666, 117.0336	13	269, 331	Apigenin *C*-pentoside *C*-hexoside III	2.06 × 10^6^	4.67%
53	18.87	447.0935	448.1006	C_21_H_20_O_11_	99.41	−0.46	−0.21	327.0515, 179.0141, 135.0447	12	268, 325	Luteolin *C*-hexoside I	2.94 × 10^5^	0.67%
54	19.5	447.093	448.1006	C_21_H_20_O_11_	91.66	0.28	0.21	327.0539, 179.0138, 135.0450	12	N.D.	Luteolin *C*-hexoside II	1.92 × 10^5^	0.44%
55	20.34	151.04	152.0473	C_8_H_8_O_3_	98.04	−0.3	0	136.0167, 92.0269	5	256	Methoxybenzoic acid I	9.28 × 10^5^	2.11%
56	20.4	163.0399	164.0473	C_9_H_8_O_3_	97.35	0.47	0.08	N.D.	6	N.D.	*p*-Coumaric acid *	1.17 × 10^5^	0.27%
57	20.46	609.1472	610.1534	C_27_H_30_O_16_	96.31	−1.41	−0.86	300.0267, 151.0033	13	255, 355	Quercetin 3-*O*-rutinoside (rutin) *	1.27 × 10^6^	2.88%
58	20.59	593.151	594.1585	C_27_H_30_O_15_	99.5	0.3	0.16	447.0897, 285.0396, 133.0281	13	267, 320	Luteolin deoxyhexoside hexoside	1.65 × 10^5^	0.37%
59	20.93	300.9988	302.00627	C_14_H_6_O_8_	94.92	0.82	0.25	N.D.	12	N.D.	Ellagic acid *	2.68 × 10^4^	0.06%
60	21.06	463.0874	464.09548	C_21_H_20_O_12_	84.99	0.25	0.12	300.9980, 151.0030	12	N.D.	Quercetin 3-*O*-β-D-galactopyranoside *	5.82 × 10^4^	0.13%
61	21.18	623.1972	624.2054	C_29_H_36_O_15_	94.2	1.52	0.95	461.1649, 387.1502, 315.1074, 297.0957, 179.0347, 161.0239, 153.0543, 135.0447, 113.0233	12	N.D.	Verbascoside	2.80 × 10^5^	0.64%
62	21.18	681.2402	682.2473	C_32_H_42_O_16_	99.38	−0.12	−0.08	519.1525, 357.1333, 179.0529,151.0382, 149.0467	12	N.D.	Pinoresinol dihexoside II	7.45 × 10^4^	0.17%
63	21.3	463.0883	464.09548	C_21_H_20_O_12_	98.82	−0.02	−0.01	301.0324, 300.0248 271.02228, 255.0276, 178.9974, 151.0027, 136.0172, 135.0447	12	250, 352	Quercetin 3-*O*-β-D-glucopyranoside *	9.70 × 10^5^	2.20%
64	21.36	447.0936	448.1006	C_21_H_20_O_11_	97.26	−0.84	−0.38	285.0402,151.0026, 133.0284	12	N.D.	Luteolin 7-*O*-β-D-glucopyranoside *	1.27 × 10^6^	2.89%
65	21.87	575.1189	576.1268	C_30_H_24_O_12_	92.55	0.51	0.3	N.D.	19	N.D.	Procyanidin A2 *	1.22 × 10^4^	0.03%
66	21.79	1017.3111	1018.3165	C_44_H_58_O_27_	96.98	−1.71	−1.74	855.2573, 693.2019, 369.0973, 323.0977, 221.0642, 219.0663, 179.0559, 161.0452, 149.0451, 143.0349	16	280	Sesaminol tetrahexoside I	1.86 × 10^5^	0.42%
67	22.27	1017.3095	1018.3165	C_44_H_58_O_27_	98.45	−0.15	−0.15	855.2556, 693.2026, 369.0971, 323.0973, 221.0682, 179.0555, 161.0459, 149.0443, 143.0342	16	280	Sesaminol tetrahexoside II	3.13 × 10^5^	0.71%
68	22.39	593.1504	594.1585	C_27_ H_30_ O_15_	96.05	0.84	0.5	N.D.	13	N.D.	Kaempferol 3-*O*-rutinoside *	7.10 × 10^4^	0.16%
69	22.52	841.2775	842.2857	C_38_H_50_O_21_	99.4	−0.04	-0.04	679.2225, 485.1504, 355.1176, 323.0978, 221.0665, 179.0548, 161.0454, 149.0450, 143.0352, 121.0288, 89.0245	14	286	Xanthoxylol trihexoside	7.38 × 10^5^	1.68%
70	22.54	163.0404	164.0473	C_9_H_8_O_3_	84.16	−1.97	-0.32	N.D.	6	284	*m*-Coumaric acid *	3.22 × 10^4^	0.07%
71	22.6	193.0505	194.0579	C_10_H_10_O_4_	99.6	0.5	0.1	178.0270, 134.0371, 119.0503	6	230, 282, 310	Ferulic acid *	5.09 × 10^5^	1.16%
72	22.73	855.2582	856.2637	C_38_H_48_O_22_	96.7	−1.2	−1.55	693.2036, 485.1494, 369.0963, 323.0999, 221.0663, 179.0556, 161.0456, 149.0446, 143.0346, 119.0348	15	278	Sesaminol trihexoside I	1.14 × 10^6^	2.59%
73	23.18	609.1822	610.1898	C_28_H_34_O_15_	98.33	0.49	0.3	447.1293, 301.0713, 259.0811, 175.0023, 151.0031	12	281	Hesperetin hexoside deoxyhexoside	2.63 × 10^5^	0.60%
74	23.2	447.0943	448.1006	C_21_H_20_O_11_	96.91	−2.07	−0.93	285.0404, 135.0452, 127.0764	12	N.D.	Kaempferol 3-*O*-β-D-glucopyranoside *	7.60 × 10^5^	1.72%
75	23.3	855.2572	856.2637	C_38_H_48_O_22_	98.3	−0.6	−0.53	693.2060, 485.1514, 369.0970, 323.0983, 221.0667, 179.0560, 161.0453, 149.0452, 143.0352, 119.0351	15	277	Sesaminol trihexoside II	5.46 × 10^5^	1.24%
76	23.49	855.2568	856.2637	C_38_H_48_O_22_	99.7	−0.3	−0.24	693.2029, 485.1508, 369.0981, 323.0980, 221.0660, 179.0563, 161.0457, 149.0451, 143.0351, 119.0349	15	287	Sesaminol trihexoside III	7.90 × 10^6^	17.94%
78	24.21	871.2521	872.2586	C_38_H_48_O_23_	94.5	−0.7	−0.61	709.1983, 691.1938, 485.1527, 385.0925, 323.0985, 221.0664, 179.0556, 161.0447, 143.0354, 137.0245, 119.0354, 89.0245	15	277	Hydroxysesamolin trihexoside	4.42 × 10^5^	1.00%
79	24.27	539.1775	540.1843	C_25_H_32_O_13_	99.11	−0.8	−0.43	377.1240, 359.1192, 333.0842, 327.0880, 275.0918, 223.0641, 209.0455, 191.0349, 179.0697, 171.0293, 161.0473, 153.0559, 127.0400	10	235, 273	*Sinapoyl-3-dehydroshikimic acid hexoside I*	4.56 × 10^6^	10.36%
80	24.24	135.0451	136.0525	C_8_H_8_O_2_	85.4	−0.17	−0.02	77.04	5	273	Methyl benzoic acid	2.07 × 10^6^	4.69%
81	24.43	679.2248	680.2319	C_32_H_40_O_16_	98.53	−0.35	−0.24	517.2748, 485.1429, 355.1176, 323.0964, 221.0661, 179.0566, 161.0453, 149.0449, 143.0342, 121.0288, 89.0244	13	282	**Xanthoxylol dihexoside**	3.27 × 10^5^	0.74%
82	24.76	539.1775	540.1843	C_25_H_32_O_13_	99.09	0.52	0.28	333.0854, 327.0889, 223.0609 209.0446, 191.0340, 171.0299, 161.0482, 153.0537, 127.0399	10	230, 277	**Sinapoyl-3-dehydroshikimic acid hexoside II**	2.46 × 10^5^	0.56%
83	24.82	161.0249	162.0322	C_9_H_6_O_3_	75.63	−3.38	−0.19	N.D.	7	N.D.	7-hydroxycoumarin (umbelliferone) *	1.60 × 10^4^	0.04%
84	24.97	151.04	152.0473	C_8_H_8_O_3_	99.76	−0.1	0	136.0167, 92.0270	5	N.D.	Methoxybenzoic acid II	2.41 × 10^5^	0.55%
85	26.45	285.0408	286.0477	C_15_H_10_O_6_	99.42	−1.2	−0.3	227.1288, 135.0450	11	287, 325	Luteolin *	8.39 × 10^5^	1.91%
86	28.32	271.0614	272.0685	C_15_H_12_O_5_	99.03	−0.7	−0.2	N.D.	10	N.D.	Naringenin *	1.68 × 10^5^	0.38%

^a^ Detected ions were [M-H]^−^. * Compounds confirmed by standards comparison; N.D., below 5 mAU or undetected due to masking by compounds with higher signal. The letter codes I, II, etc. indicate different isomers. New proposed structures are in bold., DBE: double bond equivalents. Lowest value 
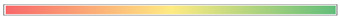
 Highest value.

**Table 2 foods-10-00298-t002:** Non-phenolic compounds characterized in SG32 oil.

Peak No.	RT (min)	Experimental *m/z* ^a^ [M-H]^−^	Theoretical Mass (M)	Molecular Formula	Error (ppm)	Error (mDa)	Score	Main Fragments	DBE	UV (nm)	Proposed Compound	Area (Response × RT)	%
1	2.65	195.0506	196.0583	C_6_H_12_O_7_	2.33	0.046	98.38	165.0398	1	N.D.	Gluconic/galactonic acid	2.56 × 10^5^	0.8
2	3.09	191.0201	192.027	C_6_H_8_O_7_	−1.72	−0.33	99.05	173.0094, 111.0089	3	N.D.	Citric acid I	2.19 × 10^6^	6.5
3	3.15	133.0142	134.0215	C_4_H_6_O_5_	0.29	0.04	98.8	115.0039	2	N.D.	Malic acid I	9.99 × 10^5^	2.9
4	3.40	133.0141	134.0215	C_4_H_6_O_5_	1.37	0.18	99.11	115.004	2	N.D.	Malic acid II	9.83 × 10^5^	2.9
5	4.02	191.0203	192.027	C_6_H_8_O_7_	−2.91	−0.56	97.87	173.0096, 111.0092	3	N.D.	Citric acid II	2.46 × 10^6^	7.3
6	4.08	128.0358	129.0426	C_5_H_7_NO_3_	−2.98	−0.38	95.55	111.0092	3	N.D.	Pyroglutamic acid I	6.80 × 10^5^	2.0
7	4.27	191.0191	192.027	C_6_H_8_O_7_	3.25	0.62	96.75	173.0081, 111.0088	3	N.D.	Citric acid III	2.82 × 10^6^	8.3
8	4.58	128.0353	129.0426	C_5_H_7_NO_3_	−0.83	−0.11	98.45	111.0092	3	N.D.	Pyroglutamic acid II	1.36 × 10^6^	4.0
10	5.01	130.0869	131.0949	C_6_H_13_NO_2_	3.76	0.49	95.92	112.9856	1	N.D.	Leucine/Isoleucine I	4.05 × 10^5^	1.2
11	5.45	117.0193	118.0266	C_4_H_6_O_4_	0	0	98.47	73.0298	2	N.D.	Succinic acid	4.22 × 10^5^	1.2
12	5.51	130.087	131.0949	C_6_H_13_NO_2_	2.59	0.34	99.11	112.9856	1	N.D.	Leucine/Isoleucine II	6.60 × 10^5^	1.9
13	5.82	180.0668	181.0745	C_9_H_11_NO_3_	1.73	0.31	95.85	163.097	5	265	Tyrosine *	6.00 × 10^5^	1.8
14	6.01	130.087	131.0949	C_6_H_13_NO_2_	2.98	0.39	85.93	112.9856	1	N.D.	Leucine/Isoleucine III	5.14 × 10^5^	1.5
15	6.01	611.1444	612.152	C_20_H_32_N_6_O_12_S_2_	0.25	0.016	98.41	306.0731, 305.0671, 128.0366	8	N.D.	Oxidized Glutathione (glutathione disulfide)	1.25 × 10^6^	3.7
17	7.75	171.0293	172.0372	C_7_H_8_O_5_	3.2	0.6	83.75	127.0402	4	230	(-)-3-dehydroshikimic acid	4.07 × 10^5^	1.2
18	7.87	191.0563	192.0634	C_7_H_12_O_6_	−0.8	−0.2	99.72	147.0665, 129.0556, 101.0608	2	N.D.	Quinic acid I	3.49 × 10^6^	10.3
19	8.18	191.0562	192.0634	C_7_H_12_O_6_	−0.3	−0.1	99.45	147.0660, 129.0556, 101.0607	2	N.D.	Quinic acid II	6.29 × 10^6^	18.5
22	9.20	164.0718	165.0790	C_9_H_11_NO_2_	−0.3	0.0	99.72	147.049	5	N.D.	Phenylalanine *	3.64 × 10^6^	10.7
23	10.61	218.1031	219.1107	C_9_H_17_NO_5_	1.33	0.3	97.8	146.0819	2	N.D.	Pantothenic acid (Vit B5)	2.09 × 10^5^	0.6
29	13.05	203.0834	204.0906	C_11_H_12_N_2_O_2_	−1.9	−0.4	86.6	142.0663, 116.0507	7	277	Tryptophan *	1.96 × 10^6^	5.8
34	14.49	175.0612	176.0685	C_7_H_12_O_5_	0.19	0.03	99.66	115.0402	2	N.D.	Isopropylmalic acid	2.48 × 10^5^	0.7
77	24.05	187.0979	188.1049	C_9_H_16_O_4_	−1.7	−0.3	99	125.097	2	N.D.	Azelaic acid	2.06 × 10^6^	6.1

^a^ Detected ions were [M-H]^−^. * Compounds confirmed by standards comparison; N.D., below 5 mAU or undetected due to masking by compound with higher signal. The letter codes I, II, etc. indicate different isomers. DBE: double bond equivalents. Lowest value 
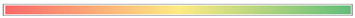
 Highest value.

## Data Availability

Part of the data presented in this study is available in Appendix A. Other data are available on request from the corresponding author.

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
