# Peer review of "Metabolic Profiling of the Oil of Sesame of the Egyptian Cultivar ‘Giza 32’ Employing LC-MS and Tandem MS-Based Untargeted Method"

_foods, 2021, doi:10.3390/foods10020298_

Round 1

Reviewer 1 Report

The manusript entitled "Metabolic Profiling of the oil of sesame of the Egyptian cultivar ‘Giza 32’ employing LC-MS and tandem MS-based untargeted
method" describes the characterization of the sesame oil metabolites through two detection methods.

In my opinion the manuscript is well written and the discussion well fundamented with several reports from the literature.

Comments:

  • Please format section 3. results and discussion. 
  • The authors used positive and negative modes for identification? Please add that information
  • Please format Table 1, once the font number is too high and the information presented in the Table becomes confusing

Author Response

Reviewer 1

Open Review

English language and style

( ) Extensive editing of English language and style required
( ) Moderate English changes required
( ) English language and style are fine/minor spell check required
(x) I don't feel qualified to judge about the English language and style

Yes

Can be improved

Must be improved

Not applicable

Does the introduction provide sufficient background and include all relevant references?

(x)

( )

( )

( )

Is the research design appropriate?

(x)

( )

( )

( )

Are the methods adequately described?

(x)

( )

( )

( )

Are the results clearly presented?

(x)

( )

( )

( )

Are the conclusions supported by the results?

(x)

( )

( )

( )

Comments and Suggestions for Authors

The manusript entitled "Metabolic Profiling of the oil of sesame of the Egyptian cultivar ‘Giza 32’ employing LC-MS and tandem MS-based untargeted
method" describes the characterization of the sesame oil metabolites through two detection methods.

In my opinion the manuscript is well written and the discussion well fundamented with several reports from the literature.

Thanks very much for your encouragement.

Comments:

  • Please format section 3. results and discussion. 

This part was formatted according to your comment. Moreover, the format of the rest of sections has been revised.

  • The authors used positive and negative modes for identification? Please add that information

Only the negative mode was used since it provided good results in our previous work on sesame cake. It is mentioned in the material and methods section point 2.3 and the results and discussion section point 3.1

  • Please format Table 1, once the font number is too high and the information presented in the Table becomes confusing

Both tables 1 and 2 were formatted.

Reviewer 2 Report

The article entitled “metabolic profiling of the oil of sesame of the Egyptian cultivar ‘Giza 32’ employing LC-MS and tandem MS-based untargeted method” deals with the identification of metabolites in a polar fraction of sesamum oil. The data generated here are useful however manuscript needs improvement .

I have the following comments for this article:

  1. Please provide the discussion of why cultivar ‘Giza 32’ was particularly selected in the present study
  2. Please provide the optimized LC-MS/MS parameters for the compounds identified using MS/MS spectrum, e.g., collision energy, Fragmentor voltage, etc.,
  3. The fragmentation pattern given in Figs. 2-5 is not clear, and why only a few fragments were generated? Can we accurately identify compounds with just a few MS/MS fragments? please provide a discussion.
  4. Table 1, define abbreviations (e.g., DBE) in the table footnotes
  5. Discussion: please provide the applications/usefulness of the data generated in the present study

Author Response

Reviewer 2

Open Review

English language and style

( ) Extensive editing of English language and style required
( ) Moderate English changes required
(x) English language and style are fine/minor spell check required
( ) I don't feel qualified to judge about the English language and style

Yes

Can be improved

Must be improved

Not applicable

Does the introduction provide sufficient background and include all relevant references?

( )

(x)

( )

( )

Is the research design appropriate?

( )

(x)

( )

( )

Are the methods adequately described?

( )

( )

(x)

( )

Are the results clearly presented?

( )

( )

(x)

( )

Are the conclusions supported by the results?

( )

(x)

( )

( )

Comments and Suggestions for Authors

The article entitled “metabolic profiling of the oil of sesame of the Egyptian cultivar ‘Giza 32’ employing LC-MS and tandem MS-based untargeted method” deals with the identification of metabolites in a polar fraction of sesamum oil. The data generated here are useful however manuscript needs improvement .

Thanks very much for your encouragement. All your comments have been considered for the revised version of the work.

I have the following comments for this article:

1. Please provide the discussion of why cultivar ‘Giza 32’ was particularly selected in the present study

We used this cultivar because of availability in Cairo market and since it was used for our previous work on sesame cake. This enabled to compare the metabolites from both parts, the case and the oil. Moreover, after literature review we could find that there is no studies dealing with untargeted metabolic profiling for the polar fraction for sesame oil in general and for this cultivar in specific.

2. Please provide the optimized LC-MS/MS parameters for the compounds identified using MS/MS spectrum, e.g., collision energy, Fragmentor voltage, etc.,

This part was mentioned in the supplementary material part.

3. The fragmentation pattern given in Figs. 2-5 is not clear, and why only a few fragments were generated? Can we accurately identify compounds with just a few MS/MS fragments? please provide a discussion.

The identification was tentative based not only on the fragmentation patterns but also in the molecular formula and information retrieved from literature. Main fragments are shown in the tables. In the figure, we highlighted major and/or characteristics ions by adding information regarding characteristic neutral losses of the phenolic compounds and the fragments that they correspond (this have now been clarified in page 3). This was done to try that readers can imagine the structure based on the, for example, functional groups like methyl, carboxyl, etc. (now explained in page 8), or basic constituents, e.g. the presence of O-hexoside or C-hexoside (now better explained in page 6, 8 and 9), caffeoyl and quinic acid moieties (page 6), etc., Also, this was done due to limit the dimensions of the figures and Tables.

4. Table 1, define abbreviations (e.g., DBE) in the table footnotes

DBE stands for double bond equivalence, it was added as a table footnote.

5. Discussion: please provide the applications/usefulness of the data generated in the present study.

According to your comment, the application/usefulness of this work has been commented in section 3.3.

Reviewer 3 Report

The present work is an original and scientifically well processed and obtained results will benefit in the scientific field of use of selective method based on RP-HPLC-DAD-QTOF-MS and tandem MS. The results are processed in the table and the attached images are clear and well arranged. References are adequate and suitable in quality and quantity.

In the article I would recommend a minor editing in the introduction - "porducing oil" - producing oil (page1, line 3), Hassan 2012 [6] - Hassan [6] - (p 2), in the paragraph on page 3 - The following databases were ...... I would consider including internet links in the References section, in tables 1 (p 10) and 2 (p 15) I propose to add units for the parameter Area and for the title - 3.3. Comparison between Sesame Seed Oil and Cake. (p 16) edit format.

Author Response

Reviewer 3

Open Review

English language and style

( ) Extensive editing of English language and style required
( ) Moderate English changes required
(x) English language and style are fine/minor spell check required
( ) I don't feel qualified to judge about the English language and style

Yes

Can be improved

Must be improved

Not applicable

Does the introduction provide sufficient background and include all relevant references?

(x)

( )

( )

( )

Is the research design appropriate?

(x)

( )

( )

( )

Are the methods adequately described?

(x)

( )

( )

( )

Are the results clearly presented?

(x)

( )

( )

( )

Are the conclusions supported by the results?

(x)

( )

( )

( )

Comments and Suggestions for Authors

The present work is an original and scientifically well processed and obtained results will benefit in the scientific field of use of selective method based on RP-HPLC-DAD-QTOF-MS and tandem MS. The results are processed in the table and the attached images are clear and well arranged. References are adequate and suitable in quality and quantity.

Thanks very much for your encouragement.

In the article I would recommend a minor editing in the introduction - "porducing oil" - producing oil (page1, line 3),

It was modified

 Hassan 2012 [6] - Hassan [6] - (p 2),

It was modified

 in the paragraph on page 3 - The following databases were ...... I would consider including internet links in the References section,

They were removed from the text and added to the references section

in tables 1 (p 10) and 2 (p 15) I propose to add units for the parameter Area

It was added (response × RT)

for the title – 3.3. Comparison between Sesame Seed Oil and Cake. (p 16) edit format.

It was modified. Moreover, the format of the rest of sections has been revised.